# The Role of Membrane Lipids in Light-Activation of *Drosophila* TRP Channels

**DOI:** 10.3390/biom12030382

**Published:** 2022-02-28

**Authors:** Rita Gutorov, Ben Katz, Elisheva Rhodes-Mordov, Rachel Zaguri, Tal Brandwine-Shemmer, Baruch Minke

**Affiliations:** Department of Medical Neurobiology, Institute for Medical Research Israel-Canada (IMRIC), Faculty of Medicine and Edmond and Lily Safra Center for Brain Sciences (ELSC), The Hebrew University, Jerusalem 91120, Israel; rita.gutorov@mail.huji.ac.il (R.G.); ben.katz@mail.huji.ac.il (B.K.); elisheva.rhodes@mail.huji.ac.il (E.R.-M.); rachel.segura@mail.huji.ac.il (R.Z.); tal.brandwine@mail.huji.ac.il (T.B.-S.)

**Keywords:** *Drosophila* TRP channel, TRPL channel, phospholipase Cβ, Diacylglycerol (DAG), Diacylglycerol kinase (DGK), poly unsaturated fatty acids (PUFAs), methyl-β-cyclodextrin, cholesterol, ergosterol

## Abstract

Transient Receptor Potential (TRP) channels constitute a large superfamily of polymodal channel proteins with diverse roles in many physiological and sensory systems that function both as ionotropic and metabotropic receptors. From the early days of TRP channel discovery, membrane lipids were suggested to play a fundamental role in channel activation and regulation. A prominent example is the *Drosophila* TRP and TRP-like (TRPL) channels, which are predominantly expressed in the visual system of *Drosophila*. Light activation of the TRP and TRPL channels, the founding members of the TRP channel superfamily, requires activation of phospholipase Cβ (PLC), which hydrolyzes phosphatidylinositol 4,5-bisphosphate (PIP_2_) into Diacylglycerol (DAG) and Inositol 1, 4,5-trisphosphate (IP_3_). However, the events required for channel gating downstream of PLC activation are still under debate and led to several hypotheses regarding the mechanisms by which lipids gate the channels. Despite many efforts, compelling evidence of the involvement of DAG accumulation, PIP_2_ depletion or IP_3_-mediated Ca^2+^ release in light activation of the TRP/TRPL channels are still lacking. Exogeneous application of poly unsaturated fatty acids (PUFAs), a product of DAG hydrolysis was demonstrated as an efficient way to activate the *Drosophila* TRP/TRPL channels. However, compelling evidence for the involvement of PUFAs in physiological light-activation of the TRP/TRPL channels is still lacking. Light-induced mechanical force generation was measured in photoreceptor cells prior to channel opening. This mechanical force depends on PLC activity, suggesting that the enzymatic activity of PLC converting PIP_2_ into DAG generates membrane tension, leading to mechanical gating of the channels. In this review, we will present the roles of membrane lipids in light activation of *Drosophila* TRP channels and present the many advantages of this model system in the exploration of TRP channel activation under physiological conditions.

## 1. Introduction

The Transient Receptor Potential (TRP) superfamily, which is conserved through evolution, consists of seven subfamilies (TRPC, TRPV, TRPM, TRPA, TRPN, TRPML, and TRPP) and its members are expressed in many cell types, including excitable as well as non-excitable cells [1]. These channels participate in many sensory modalities (e.g., vision, taste, temperature, pain, and pheromone detection) and they either open directly in response to ligands or physical stimuli (e.g., temperature, osmotic pressure) or open indirectly, downstream of receptor activation (e.g., Rhodopsin, Histamine, and Bradykinin [2,3]) and the inositol-lipid signaling cascade. In addition to the wide variety of activation mechanisms, the activity of TRP channels is modulated by numerous factors, including the lipid environment in which the channel is embedded, posttranslational modifications such as phosphorylation, nitrosylation and glycosylation, ligand biding such as Ca^2+^ and ATP and by interaction of the channels with different binding partners [4]. Despite many efforts over the years, the gating mechanism of the founding members of the TRP channel superfamily, the *Drosophila* TRP and TRP-like (TRPL) channels is still elusive.

Visual systems of animals are characterized by high sensitivity to light and this property is obtained by a high expression level of the signaling components within the retinal cells. The retina of *Drosophila* is an excellent example of this principle as revealed by expression of huge amounts of signaling components and by achieving the ultimate sensitivity to light–single photon detection. *Drosophila* utilizes the phosphoinositide cascade for phototransduction with TRP channels as its target [2,3]. Hence, the *Drosophila* retina is a unique tissue with regard to TRP channels, as most tissues do not express high amounts of TRP channels. This TRP-enriched tissue enables robust biochemical analysis combined with the power of the *Drosophila* molecular genetics and the accuracy of light activation and constitutes a valuable preparation for investigating the complex mechanism of TRP channel gating under physiological conditions. Despite having many advantages, the *Drosophila* photoreceptor preparation still has several limitations, including the polarity of the *Drosophila* photoreceptor cells and the dense membrane of the signaling compartment (rhabdomere), which limits pipette accessibility to the channels, making it difficult to directly determine the single channel properties under physiological conditions ([5], but see below).

Lipid regulation of channels is mostly thought as ligand-protein interaction with a defined binding site at the channel surface. However, recent in-silico lipid docking analysis using solved channel structures suggests that some lipid binding sites enable binding of multiple lipid “poses” [6]. Another interesting theory developed for the gating mechanism of *Drosophila* TRP/TRPL channels proposed that the TRP/TRPL channels are mechanically gated. Accordingly, the measured force generated by changes in membrane lipid packing and the generation of membrane tension during the enzymatic activity of PLC and the conversion of PIP_2_ into DAG, gate the mechanical sensitive TRP/TRPL channels. This theory emphasizes that the lipid environment in which the channels are embedded is crucial for their performance.

In the present review we summarize the evidence supporting the involvement of lipids in light-activation of the TRP/TRPL channels. Moreover, we will discuss conflicting and missing data required for elucidating the mechanism underlying lipid-activation of the TRP/TRPL channels.

## 2. The Involvement of the Inositol-Lipid Signaling in TRP Channel Activation

One of the most important contributions of the *Drosophila* retinal preparation to the study of TRP channels in general and of the TRPC channels subfamily in particular is the discovery that Gq-mediated phospholipase Cβ (PLC) has an essential role in physiological activation of the TRP channels (Figure 1). Evidence for possible participation of PLC in *Drosophila* phototransduction arose from initial biochemical studies, showing that highly reduced PLC enzymatic activity was found in the no receptor potential A (*norpA*) mutants [7,8,9]. Further detailed functional evidence for a light-dependent G_q_α-mediated PLC activity in fly photoreceptors came from combined biochemical and electrophysiological experiments. These experiments revealed coupling of photoexcited rhodopsin to phosphatidylinositol 4,5-bisphosphate (PIP_2_) hydrolysis [10]. Furthermore, using temperature-sensitive *norpA* mutant allele [11], correlation between light-activated PIP_2_ hydrolysis and the electrical response to light was demonstrated [12]. The findings of the biochemical and electrophysiological studies were strongly supported by a genetic study, resulting in the isolation and analysis of the *Drosophila* gene, *norpA,* which was found to encode a β-class PLC, predominately expressed in the rhabdomeres and has high amino acid homology to a PLC extracted from bovine brains [13]. The *norpA* mutant thus provides essential evidence for the critical role of PLC and the inositol-lipid signaling in fly phototransduction, by showing that no light excitation takes place in the absence of a functional PLC. However, the events required for light excitation downstream of PLC activation remain unresolved.

A reduction in the levels of PLC in mutant flies affect the amplitude and activation kinetics of the light response [15], but surprisingly also induce the slow response termination. This slow response termination of most *norpA* mutant alleles had no explanation for many years. Later biochemical and physiological studies conducted in *Drosophila* revealed the requirement for PLC in GTPase-activating proteins (GAP) activity in vivo [16], suggesting that the PLC enzyme also acts as a GAP [17]. The virtually complete dependence of GAP activity on PLC provides an efficient mechanism for ensuring the one photon, one single photon response (quantum bump) relationship [18], which is critical for the fidelity of phototransduction at dim light [16]. PLC activity is known to be regulated by Ca^2+^ [19]. It has been shown that Ca^2+^ is bound to the catalytic site of PLC and is required as a co-factor for the catalytic reaction [20]. These studies showed that the positive charge of Ca^2+^ is used to counterbalance local negative charges formed in the active site during the course of the catalytic reaction. Accordingly, Ca^2+^ electrostatically stabilizes both the substrate and the transition state, thus providing a 2-fold contribution to lower the energy of the enzymatic reaction [21]. In *Drosophila*, both in vitro [22] and in vivo [23] measurements revealed Ca^2+^ dependence of PLC activity. This activity shows a bell-shaped dependence of PLC catalytic activity on Ca^2+^ concentration ([Ca^2+^]), with maximal basal activity in the range of 10^−7^–10^−5^ M [Ca^2+^]. This complex dependency affects both excitation and adaptation of the photoresponse. While at physiological conditions cellular Ca^2+^ levels are sufficient for the activation of PLC, it was demonstrated that highly reduced Ca^2+^ levels eliminated excitation completely [24]. Furthermore, it was shown that during the light response, rhabdomeric Ca^2+^ concentration can reach mM levels [25,26]. At this high Ca^2+^ level, PLC activity is attenuated, raising the possibility of physiological relevance [23]. Later studies have shown that the inhibition of PLC activity by high Ca^2+^ concentration participates in the mechanism of fast light adaptation and prevents depletion of membrane PIP_2_. Accordingly, the high Ca^2+^ concentration, reached at the peak response, attenuates PLC activity, preventing PIP_2_ depletion and adapting the cells by reducing excitation [27]. The transient response phenotype observed in *trp* mutants was explained by this same mechanism [28]. Accordingly, the reduced Ca^2+^ permeability in the *trp* mutant, results in reduced Ca^2+^ influx during illumination through the low Ca^2+^ permeable TRPL channels. This reduced Ca^2+^ influx becomes insufficient to attenuate PLC activity, thereby depleting the pool of PIP_2_ and causing the attenuation of the light response [28,29]. Indeed, the response to light of WT flies under low Ca^2+^ conditions also showed a *trp* phenotype [30,31]. The *trp* phenotype was also observed in null *rdgB* mutants in which the PI-cycle (Figure 2) is defective and PIP_2_ depletion is expected after intense light illumination ([32], but see [33]).

Initially, models of light-activated TRP/TRPL channels assumed that the IP_3_ branch of the inositol-lipid signaling leads to IP_3_-induced Ca^2+^ release from intracellular stores (Figure 1) mediating light-induced TRP/TRPL channels opening [34,35], whereby the released Ca^2+^ functions as a second messenger of excitation acting either in a store depletion mechanism (SOC, [36]) or as a receptor-activated channel (ROC, [37]). However, in *Drosophila* these mechanisms were not supported by experimental evidence [38,39,40] and therefore were abandoned and the lipid branch of PLC-dependent signaling was adopted [41,42]. Recently, a rapid light-induced release of Ca^2+^ from the ER by the Na^+^/Ca^2+^ exchanger (CALX, [41]) was demonstrated (Figure 1). However, This CALX-dependent Ca^2+^ release was too slow to be involved in channel activation and there is no information on ER Na^+^ levels. Thus, the physiological role of this mechanism is still unknown.

## 3. Lipid Composition of the *Drosophila* Head/Retina and the Effects of Its Modification

There is mounting evidence that lipids have profound effects on the activity of ion channels in general and on members of the TRPC subfamily in particular [44]. Lipids of *Drosophila* head membranes have been analyzed using thin layer chromatography, gas liquid chromatography [45,46], and nano-electrospray ionization tandem mass spectrometry (ESI-MS/MS) [47]. These studies revealed the presence of ~50% phosphatidyl-ethanolamine (PE), ~25% phosphatidyl-choline (PC), and ~12% phosphatidyl-inositol (PI). Contrary to vertebrate photoreceptors, long chain polyunsaturated fatty acids (more than 18 carbons) were not detected in *Drosophila* phospholipids [46]. In addition to phospholipids, *Drosophila* head membranes contain sphingolipids and, instead of cholesterol, ergosterol [47,48]. The ergosterol content of *Drosophila* head membranes was determined by ESI-MS/MS and revealed ~26 nmol ergosterol per mg of membrane proteins [47]. The sphingolipids are composed of a tetradeca-4-sphingenine and a saturated fatty acid [48]. Since these studies used membranes of entire *Drosophila* heads, the obtained quantitative results may not mirror the exact lipid composition of photoreceptor membranes.

Lipidomic analysis of fly heads raised on a yeast-free diet showed that the lack of Poly Unsaturated Fatty Acids (PUFAs) in this diet was reflected in marked differences in fatty acids composition of all the major phospholipids in the heads. In particular, while mono-saturated lipid species were little affected, polyunsaturated (with 3 or more double bonds) were greatly reduced in all classes of phospholipids. Interestingly PIP_2_, the PLC substrate in heads of flies reared on normal diet was in order of abundance 36:2 > 36:3 > 34:2 = 36:4 > 34:1 (carbon number: number of double bonds). In flies raised on the yeast-free diet, PUFAs of 36:4 and 36:3 were almost eliminated and 34:1 increased accordingly. The results obtained from fly heads were compared to results obtained from dissected retinae and a closely overlapping profile was observed [49]. To conclude, in flies reared on yeast-free diet lacking PUFAs, mass spectrometry showed that the proportion of polyunsaturated phospholipids was seven-fold reduced (from 38 to 5%) but rescued by adding a single species PUFA to the diet. The yeast free diet caused a 2- to 3-fold increase in light response latency and time to peak, but had no effect on the waveform of the single photon responses (quantum bumps). Interestingly, in flies raised on yeast-free diet a reduction in light sensitivity of ~10-folds was observed in WT flies under low Ca^2+^ conditions and in *trp* mutant flies in which Ca^2+^ influx is reduced. It is important to note that this diet had virtually no effect on PLC activity [49]. The effect of the membrane phospholipids manipulation under a yeast-free diet is expected to change the stiffness and elasticity of the membrane. Hence, the authors suggested that a 2- to 3-fold increase in response latency, which appeared to be mediated downstream of PLC and was correlated with a slowing of the light-induced contractions in accordance with mechanical gating of the channels (see below).

## 4. Evidence for Lipids Action as Second Messengers

Many mammalian TRP channels can be characterized as ionotropic receptors since they can be activated directly under physiological conditions by physical stimuli such as heat, cold, mechanical, or by natural chemicals such as capsaicin, menthol, or mustard oil [50]. One of the key features of presumably all members of the TRPC subfamily is their indirect physiological activation via PLCβ-mediated signal transduction cascades. Since PLC activity is essential for activation of the TRP/TRPL channels, recycling its substrate, PIP_2_, is essential for proper light response. PLC catalyzes hydrolysis of the membrane phospholipid PIP_2_ into water soluble IP_3_ and membrane-bound diacylglycerol (DAG, [51]). This enzymatic reaction is enhanced by illumination and initiates a cyclic enzymatic pathway called the PI cycle (Figure 2). Many enzymatic components of the complex PI cycle were discovered using genetic dissection by screening for mutations and utilizing the power of the *Drosophila* molecular genetics [52]. Following PLC activation, the phospholipid branch of the PI cycle begins by DAG transport through endocytosis to the endoplasmic reticulum designated submicrovillar cisternae (SMC, but see below). Subsequently, DAG is inactivated by phosphorylation and converted into phosphatidic acid (PA, [53,54] via DAG kinase (DGK), encoded by the retinal degeneration A (*rdgA*) gene that was discovered as a mutant causing rapid retinal degeneration in the dark [53,54]. Then, CDP-DAG synthase enzyme encoded by the *cds* gene [55] produces CDP-DAG from PA (Figure 2). Subsequently, CDP-DAG is converted into phosphatidylinositol (PI), which is transferred back to the microvillar membrane, by the PI transfer protein (PITP), encoded by the retinal degeneration B (*rdgB* gene [56]) located to the SMC. This mutant was discovered as a mutant causing rapid light-induced retinal degeneration [57,58,59]. PIP and PIP_2_ are produced at the microvillar membrane by PI kinase and PIP kinase, respectively. PA can be reconverted back to DAG by lipid phosphate phosphohydrolase, LPP, also designated phosphatidic acid phosphatase, PAP, encoded by the *laza* gene [60,61] or produced from phosphatidylcholine (PC) by phospholipase D (PLD), encoded by the *pld* gene [62,63]. Light-regulated PLD activity is required to maintain PA levels during illumination and support the maintenance of apical membrane (rhabdomere) size. Thus, PLD is a key regulator of plasma membrane turnover during receptor activation and signaling in photoreceptors [63], while the PA level is tightly regulated by phosphorylation-dephosphorylation enzymatic reactions carried out by the LPP phosphatase (laza), and DAG kinase (*rdgA*), both localized to the ER (Figure 2, but see below). PLD also works in coordination with retromer function and ADP-ribosylation factor 1 (Arf1) activity to regulate apical membrane size during illumination [63].

Mutations in several enzymes of the PI cycle result in retinal degeneration. For example, mutations in RDGA result in a light-independent retinal degeneration, while mutations in RDGB and CDS result in light-dependent degeneration (Figure 2). The mechanism underlying light-dependent degeneration of the *rdgB* and *cds* mutants is unknown. However, for the light-independent retinal degeneration of the *rdgA* mutant flies, a mechanism for the degeneration was proposed. Accordingly, the degeneration occurs due to a sustained Ca^2+^ influx through the light-activated TRP and TRPL channels. Since mutations affecting proteins of the PI cycle downstream to PLC activation lead to retinal degeneration due to toxic increasing cellular Ca^2+^ (*rdgA*, [64]), or possibly due to PIP_2_ depletion [28], the PI cycle is relevant for understanding phototransduction and TRP channel activation. A detailed analysis of the rdgA mutant encoding DAG kinase established the importance of the DAG branch in channel activation. In addition to light independent retinal degeneration, this mutant shows constitutive activity of the light activated channels, while a partial rescue of the degeneration was observed, when *rdgA* was crossed into a *trp* mutant background (*rdgA^1^;trp^P343^*, [64]). Furthermore, it has been shown that light response is partially rescued in hypomorphic mutant flies with severe reduction in light sensitivity, such as *norpA^P16^* and *G_q_α^1^* on the background of *rdgA* (*norpA^P16^*, *rdgA^1^* and *rdgA^1^*/+;*G_q_α^1^*), while rdgA amplifies the light response, thus supporting the hypothesis that DAG is involved in channel activation [65]. This hypothesis is consistent with a recent observation reporting that the human TRPC5 channel exhibits a binding site for DAG near the pore region, commensurate with a key role of DAG for TRPC5 channel activation [66,67]. Omission of ATP from the whole-cell recording pipette combined with application of metabolic inhibitors activate the TRP and TRPL channels in the dark [68,69,70] (Figure 3). This dark activation of the channel [71] by metabolic inhibition [68] was effectively mimicked by the *rdgA* mutation in DAG kinase (DGK, [64]), suggesting that ATP depletion mainly inhibits DGK activity. It furthermore suggests that DGK normally suppresses spontaneous dark activity of PLC-mediated production of DAG [70]. Indeed, a considerable spontaneous Gqα activity inducing spontaneous PLC activity in the dark was observed in *Drosophila* photoreceptors [29,72]. Accordingly, relatively large currents were reversibly induced within only a few seconds of application of mitochondrial uncouplers (either 10 μm CCCP or 0.1 mm DNP) to WT *Drosophila* photoreceptors in recordings made soon after establishing the whole-cell configuration (Figure 3). The light-sensitive channels were invariably activated by DNP or CCCP either with or without ATP in the electrode, although activation was quicker and not indefinitely reversible when no ATP was included. However, when DNP or CCCP were applied after the spontaneous dark current was eliminated by strong Gqα or PLC hypomorph mutants, no current activation by metabolic inhibition was observed, indicating that the dark current induced by metabolic inhibition is generated by Gqα-mediated PLC activity [70]. The most obvious interpretation of these results is that metabolic inhibition activates the TRP/TRPL channels via the same molecular mechanism by which light activates these channels, namely by generation of DAG that, in a still unclear manner, is involved in channel activation. Several lines of evidence challenge this hypothesis. (a) Application of DAG to intact ommatidia or to tissue culture cells expressing TRPL did not activate the channels (our unpublished data, Hardie personal communication), while application of DAG to TRPC3 channels expressed in these tissue culture cells did activate the TRPC3 channels serving as a positive control [73]. Nevertheless, application of DAG analogues 1-oleoyl-2-acetyl-sn-glycerol (OAG) at low concentration (2 µM) to inside-out patches excised from the microvilli of dissociated *Drosophila* ommatidia resulted in activation of the TRP and TRPL [74,75]. However, channels activation was slower by 3 orders of magnitude (~60 s after application, (Delgado and Bacigalupo, 2009)) compared to the ms fast light activation [29]. (b) Immuno-gold EM localization of DAG kinase (the *rdgA* gene product) was confined to the smooth endoplasmic reticulum (SMC [54]), a relatively distant cellular compartment from the transduction machinery and this localization was consistent with detailed biochemical studies of the PI cycle using *Drosophila* mutants [52,63] making DGK localization inconsistent with controlling a second messenger level. However, a recent immunofluorescence localization showed that DGK is localized to the rhabdomeres [76], while biochemical assays revealed a light-induced increase of DAG in the *Drosophila* retina [74]. Thus, the role of DAG as a possible light-generated second messenger of excitation remains unresolved.

## 5. PUFAs Activation of TRP and TRPL Channels in the Dark

One of the striking findings of Hardie and colleagues in studies of *Drosophila* photoreceptors was the demonstration that PUFAs such as linolenic or linoleic acid (LA) are potent activators of the TRP/TRPL channels in the dark. This robust, but still slower activation relative to light activation was demonstrated in both photoreceptor cells [77,78] and in cultured S2 cells ([77,78], Figure 4) and HEK cells [73] heterologously expressing TRPL channels. Direct activation of the channels by PUFAs without the need for PLC was demonstrated by showing that PUFAs can activate blind *Drosophila* mutants virtually lacking G_q_α [77] or PLC [77,78] in a similar manner to activation of WT flies (Figure 5). In cultured S2 cells heterologously expressing TRPL channels, PUFA activation of the channels was not accompanied by PIP_2_ hydrolysis, as demonstrated by the lack of fluorescence translocation from the plasma membrane to the cytosol of GFP-PH_PLCδ1_ domain. Activation of the muscarinic receptor, which is coupled to PLC induced fluorescence translocation to the cytosol, served as a positive control ([73], Figure 5).

Most of the studies on PUFA activation of the *Drosophila* channels have been carried out in tissue culture cells. We and others have performed a significant part of the research on TRPL channels in cell culture expression systems with the aim of gaining insight into channel activation mechanism directly by single channel analysis. Extensive effort was devoted to express functional *Drosophila* TRP channels in tissue culture cells with no success. The few studies reporting functional heterologous TRP expression failed to capture the typical physiological properties of TRP as illustrated in whole cell recordings of the native photoreceptor cells ([5,79] see Figure 3 and Figure 4) and showed linear leak currents [80,81,82]. In contrast, heterologous expression of the TRPL channels accurately captured the physiological response of the native photoreceptor cells [83,84] (Figure 4). The use of expression systems for TRPL channel research has several advantages: (1) the channels are, for the most part, easily expressed in a functional manner. (2) Biophysical properties such as conductance, mean open time, and permeability are readily obtained [85]. These rigorous studies revealed that the typical outward rectification of the TRPL channels [84] results from a voltage dependent divalent open channel block mechanism [78,85]. Interestingly, application of PUFA (e.g., linoleic acid, LA) removed this divalent open channel block in the presence of divalent cations by a still unclear voltage independent mechanism (Figure 4). In general, it is important to compare the activity of a specific TRP channel in a native system with that obtained from an expression system. Indeed, many of the physiological properties observed in the photoreceptors were readily obtained and verified in expression systems. Functional expression of the TRPL channels was shown in a variety of cells including *Spodoptera frugiperda* 9 (Sf9; [86,87], Schneider 2 (S2; [77,85], and COS [88] cells and resulted in constitutively active channels, whereas the expression of the channels in Human Embryonic Kidney (HEK) cells resulted in channels that are in their closed state [88,89], mimicking the state of the TRPL channels in photoreceptor cells, in the dark. The constitutive activity of the expressed TRPL channel can be further enhanced by activation of the endogenous transduction cascade and might be related to the gating of this channel. However, it is still unclear what leads to this spontaneous activity of the TRPL channel in these expression systems, which might be important for understanding channel activation. Constitutive activity of channels can be attributed to the direct activation of the channel, for example via the lipid composition of the plasma membrane [90], or to the constitutive activation of a known or unknown element upstream to the channel. In this respect, the *Drosophila* TRPL channel expressed in S2 cells might be affected by constitutively active endogenous phospholipase C (PLC). However, PLC inhibition (with U-73122 or by a PLC mutation) did not abolish the constitutive activity of TRPL expressed in S2 cells. In addition, high expression levels of the channel also result in deregulation of the channels and may result in constitutive activity. The observation that the TRP/TRPL channels readily undergo a constitutively active state both in tissue culture cells and in vivo (e.g., spontaneously, or by DAG accumulation, respectively) may suggest that the threshold for channel activation is low, thereby having an important implication on photomechanical activation of the channels (see below).

A question arises as to whether PUFA activation of the channels in the photoreceptor cells reflects the physiological mechanism of channel activation. Supporting evidence was provided by isolation of the Inactivation No Afterpotential E (*inaE)* mutant by Pak and colleagues [91]. The *inaE* gene was identified as encoding a homologue of the mammalian *sn-1* type DAG lipase, which, rather than PUFAs, releases mono-acyl glycerol (MAGs) and, are at best, weak and slowly acting channel agonists when applied exogenously (Hardie, unpublished results, see [49]). Furthermore, the *inaE* gene product immunolocalizes to the cell body with occasional puncta in the rhabdomere [91]. Mutant flies, expressing low levels of the *inaE* gene product, have an abnormal light response, while the activation of the light sensitive channels was not prevented [91]. The discovery of the *InaE* gene was an important step in the endeavor to elucidate lipids regulation of the channels (see reviews by and [52]). However, for PUFA generation, either an sn-2 DAG lipase or an additional enzyme (MAG lipase) would be required, but there is no evidence of either in *Drosophila* photoreceptors and there is also no evidence that PUFAs are even generated in response to illumination [74].

## 6. Photomechanical Gating of the TRP/TRPL Channels

Many studies have shown that membrane-protein function is regulated by the composition of the lipid bilayer in which the proteins are embedded. The commonality of the changes in protein function by changes in their lipid environment suggests an underlying physical mechanism, and at least some of the changes are caused by altered bilayer physical properties [92]. Although TRP/TRPL channel activation by PUFA is most likely not a physiological mechanism, it provides an important insight into mechanisms by which membrane lipid modulation by a variety of PUFAs causes dramatic effects on TRP/TRPL channel gating and its properties. Accordingly, we previously showed that removal of open channel block (OCB) from TRP/TRPL channels by PUFA resulted from an increased flow rate of the blocking divalent cations through the channel pore [78]. Modulation of the interface of the channel and its surrounding membrane lipids might underlie the increase in the flow rate of the blocking cations and the removal of the OCB. We applied various methods to modify membrane lipid properties around the TRPL channel, including: stretch of the plasma membrane by hypoosmotic solutions, sequestration of PIP_2_ by polylysine, and application of various lipids. Alternatively, we blocked PUFA action by the GsMTx-4 tarantula toxin, a specific inhibitor of mechanosensitive channels, which acts on the channel-membrane lipid interface [93,94]. These results thus suggested that lipids do not affect the TRPL channel as second messengers but rather as modifiers of membrane lipid-channel interactions [78].

Hardie and Franze suggested that light-activated PLC, which hydrolyzes membrane PIP_2_ and generates DAG activates the TRP/TRPL channels by change in membrane tension. This membrane tension results from the conversion of PIP_2_ with a large hydrophilic head group into DAG, having a minute hydrophilic head group, causing changes in membrane lipid packing. This conclusion followed from a dramatic observation that light caused a fast contraction in the fly photoreceptors. They quantitated the light induced movement with atomic force microscopy and show that it reaches a peak of ∼400 nm, with an onset that precedes the onset of the light induced current. This photoreceptor contraction depended on the enzymatic activity of PLC that was blocked by the *norpA* mutation, but not on TRP/TRPL channel activity [95]. To support their hypothesis, Hardie and Franze showed that light can activate a mechanically gated ion channel—gramicidin—when inserted in the photoreceptor plasma membrane. To show that membrane tension gates the TRP/TRPL channels, they exposed the photoreceptors to hypo-osmotic solutions, anticipating that it will exert a similar effect as change in membrane lipid packing similar to PIP_2_ hydrolysis. In addition, similar to the results reported by Parnas et al. [78], hypo-osmotic solutions, though not sufficient to activate the TRPL/TRP channels, potentiated channels openings. Consistent with the interpretation that the generation of membrane tension is required for channels gating, they applied several cationic amphipaths that affected the light induced current in the expected direction. The above studies support the interesting possibility that specific changes in the physical properties of the lipid bilayer generating changes in protein-lipid interaction and an increase in membrane tension constitute a mechanical stimulus that opens TRP/TRPL channels during phototransduction. Furthermore, perhaps because of low activation threshold of TRP channels, changes in membrane tension induced by enzymatic reaction may be sufficient to gate the channels by change in membrane lipid packing, a mechanism that has not been anticipated to have any role in the mechano-sensitivity of these channels.

## 7. Lipid Rafts and Modulation of TRPL Channel Activity by Cholesterol

Lipid rafts are membrane microdomains rich with sterols, sphingolipids, and specific proteins. Lipid rafts have been found in all types of cells. It has been generally agreed that lipid rafts generate signaling platforms by assembling in close proximity to different signaling molecules for the optimal function of signal transduction cascades [96]. Many signaling proteins including ion channels have been found to localize in lipid rafts, while the association with rafts was required for the regulation of some integral membrane protein activities [97,98]. The actions of lipid rafts are often associated with the effect of cholesterol on membrane structure or directly on membrane proteins (see below). Channels have been found in detergent-resistant membrane (DRM) fractions, indicating the inclusion of channels in lipid rafts [99].

A study in *Drosophila* revealed dynamic organization of signaling components in DRM of *Drosophila* photoreceptors. It was found that the *Drosophila*-specific INAD scaffold protein [100,101] and its target proteins undergo light-induced recruitment to DRM rafts [102]. Reduction of the *Drosophila* sterol, ergosterol (see below), a key component of lipid rafts in *Drosophila*, resulted in a loss of INAD-signaling complexes associated with DRM fractions. Genetic analysis demonstrated that translocation of INAD-signaling complexes to DRM rafts required activation of the entire phototransduction cascade, while constitutive activation of the light-activated channels resulted in recruitment of complexes to DRM rafts in the dark. Mutations affecting INAD and TRP showed that INAD–TRP interaction is required for translocation of components to DRM rafts [102].

Unlike mammals, insects are unable to synthesize sterols [103] and *Drosophila melanogaster* is totally dependent on exogenously provided sterols, in the form of ergosterol obtained from a yeast diet [102,104]. Nevertheless, several enzymes of *Drosophila* do not differentiate between endogenous ergosterol and exogenous cholesterol [105]. Many studies have shown the involvement of cholesterol in the modulation of ion channels function. An efficient method to modulate the content of plasma membrane sterols is by methyl-β-cyclodextrin (MβCD), which is a cyclic oligosaccharide [106,107]. The β -cyclodextrins (7 glucose units) have high affinity for encapsulating both cholesterol and ergosterol [107]. MβCD is quite specific, allowing enrichment or a relatively rapid sequestration of cholesterol/ergosterol from living cells. Sterol-saturated MβCD is efficient as a sterol donor. The degree of cholesterol enrichment is between ~30% to ~3-fold, according to the type of cell [108,109]. When cells are incubated with high concentration of “empty” MβCD (5–10 mM) for hours (>2 h), 80–90% of total cellular cholesterol can be sequestered [108,110]. The amount of cholesterol sequestration from different cell types is a highly variable parameter [109,110,111,112]. Cholesterol sequestration leads to a dis-association of proteins from lipid rafts [113,114,115] and to a decrease in the clustering of raft-associated molecules [116]. It was shown that βCDs sequestered cholesterol from both cholesterol-rich and cholesterol-poor membrane domains [117,118,119,120]. In silico analysis revealed that cholesterol adopts multiple poses in a “cloud”, rather than occupying a single conformation at a highly flexible binding site of the channel [6], while the high structural similarities between cholesterol and ergosterol suggest that this is also likely to be the case for ergosterol. It was also suggested that cholesterol sequestration leads to perturbation of specific lipid environments. Previous observations have suggested reversible targeting of *Drosophila* [102] and mammalian TRPC channels to the cholesterol-rich membrane environment of lipid rafts. This led to the suggestion that the relatively fast (many seconds) inhibitory effects of cholesterol depletion by MβCD on mammalian TRPC channel activity may result from disruption of lipid raft architecture [44]. When *Drosophila* S2 cells heterologously expressing TRPL were perfused with MβCD, the TRPL-dependent current was rapidly abolished in less than 100 s (Figure 6 [121]), which corresponds with the fast kinetic phase of cholesterol sequestration experiments in cells [122]. Modulation of TRP channels function by direct binding of cholesterol to the channels was suggested for mammalian TRPV1 channel [123]. Cholesterol–protein interactions have been extensively studied and specific sterol recognition elements such as the cholesterol recognition amino acid consensus or cholesterol binding pockets have been demonstrated in ion channels [124]. However, these motifs create a bias, compelling researchers to focus solely on these sequences instead of performing a bias-free analyses. Indeed, recent studies on Kir 2.2 channels have shown that these motifs do not always reflect the real interactions of cholesterol with the channels [6].

In recent experiments we sequestered cholesterol by MβCD from the plasma membrane of S2 cells stably expressing TRPL. The constitutively active expressed TRPL channels (Figure 4), which showed a current-voltage relationship (I–V curves) with a marked TRPL dependent current at +80 mV were completely suppressed by the sequestration of cholesterol from the plasma membrane by MβCD. When the constitutive activity was suppressed, application of linoleic acid, the potent activator of TRP and TRPL channels [77,78] enhanced channel opening (data not shown). We also examined the effect of ergosterol sequestration from the plasma membrane of *Drosophila* photoreceptors and found a dramatic effect of ergosterol sequestration on the ability of the channels to open (data not shown). 

## 8. Concluding Remarks

Despite many efforts over the years, the gating mechanism of the *Drosophila* TRP/TRPL channels still poses a long-standing enigma. The findings that mammalian TRPC channels can be activated by exogeneous application of DAG together with the solved atomic structures of mammalian TRPC channels by cryo-EM constitute major progress towards understanding TRPC channel gating. Nevertheless, *Drosophila* retina is a highly valued preparation for investigating the roles of TRP/TRPL channels under physiological conditions. This is due to their high expression levels and their known functional role as the light-activated channels. There remains no comparable preparation for the mammalian TRPC channels that contains a TRP-enriched tissue combined with the power of the *Drosophila* molecular genetics and the accuracy of light activation. Moreover, studies of *Drosophila* phototransduction revealed that under physiological conditions, *Drosophila* TRP operates as an essential part of a multimolecular signaling complex. There are indications that mammalian TRPC channels also operate in a similar manner, but it is difficult to study these mechanisms in the native mammalian systems because of the scarcity of these signaling proteins in the native systems.

Crucial unsolved questions in *Drosophila* photoreceptor physiology are related to the mechanism by which DAG accumulation following DGK inhibition activates the channels. In particular, why exogeneous application of DAG to an excised signaling membrane activates the channels in orders of magnitude slower than light activation in the intact photoreceptor cell. The use of photoactivated DAG analogues [125] may help in solving this question.

An available *Drosophila* mutant in which the TRP channels are constitutively active (i.e., the *trp^P365^* mutant) is a highly valuable tool for investigating TRP gating in the future, which can be used together with the solved atomic structure of mammalian TRPCs. In addition, even if PUFA is not the native second messenger of excitation, its robust effect of opening the TRP/TRPL channels in the dark in vivo and TRPL in tissue culture cells can be studied in detail and provide useful information on channel gating.

Finally, the photomechanical gating mechanism of TRP/TRPL channels that was put forward by Hardie and colleagues needs to be revisited. According to this model and the supporting evidence, light-induced PIP_2_ hydrolysis by PLC converts a phospholipid with a large hydrophilic head group (PIP_2_) into DAG with a minute hydrophilic head group. This enzymatic reaction leads to a generation of a measurable force that may open the channels mechanically.

## Figures and Tables

**Figure 1 biomolecules-12-00382-f001:**
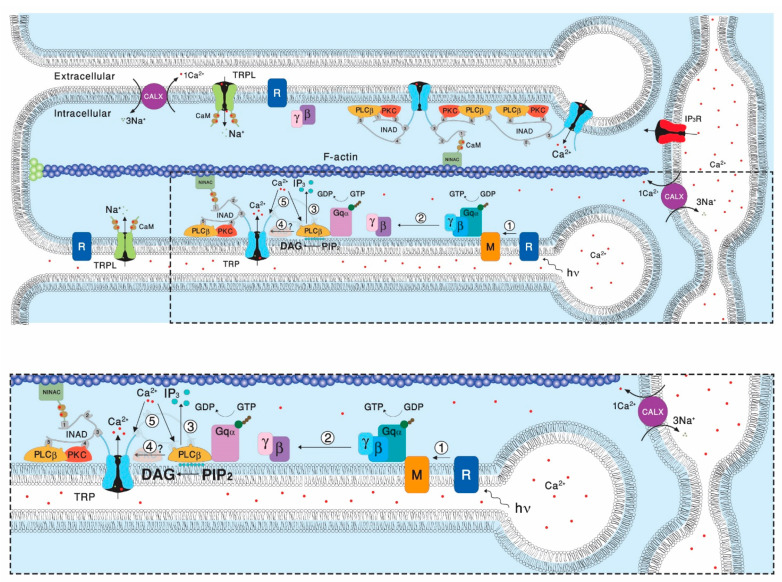
The phosphoinositide cascade of vision. A diagram showing the molecular components of the signal transduction cascade of *Drosophila*: **1.** Upon absorption of a photon (hν), rhodopsin (R) is converted into metarhodopsin (M). **2**. The R to M photoconversion leads to the activation of heterotrimeric G protein (Gqαβγ) by promoting the GDP to GTP exchange. **3**. The GTP-bound Gqα, in turn leads to activation of phospholipase Cβ (PLCβ), which hydrolyzes PIP_2_ into the soluble IP_3_ and the membrane bound DAG. **4**. PLCβ in a still unclear way activates the TRP and TRPL channel, leading to an increase in microvillar Ca^2+^ concentration. **5**. The increased Ca^2+^ concentration feeds back and negatively regulate both PLC and TRP channels activities. Elevation of DAG and Ca^2+^ promotes eye-specific protein kinase C (PKC) activity, which regulates channel activity. PLCβ, PKC, and the TRP ion channel form a supramolecular complex with the scaffolding protein INAD, which is bound to the F-actin cytoskeleton via the NINAC protein. Ca^2+^ level in the microvilli is also regulated by the Na^+^-Ca^2+^ exchanger, CALX. The diagram at the bottom is an amplification of the box marked by dotted lines in the upper diagram. (Modified from [14]).

**Figure 2 biomolecules-12-00382-f002:**
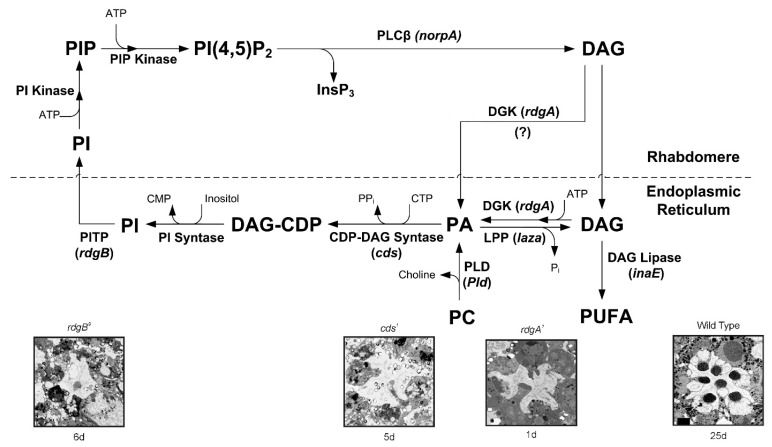
The phosphoinositide cycle in *Drosophila* photoreceptors. In the phototransduction cascade, light triggers the activation of PLCβ. This catalyzes hydrolysis of the membrane PIP_2_ into water-soluble InsP_3_ and membrane-bound DAG. The continuous functionality of the photoreceptors during illumination is maintained by rapid regeneration of PIP_2_ in a cyclic enzymatic pathway (the PI cycle). DAG is transported by endocytosis to the endoplasmic reticulum (SMC) and inactivated by phosphorylation into PA via DGK. DGK may also be localized in the rhabdomeres. PA is converted to CDP–DAG via CDP-DAG syntase (encoded by the *cds* gene). PA can be converted back to DAG by lipid phosphate phosphohydrolase (LPP; also designated phosphatidic acid phosphatase (PAP) encoded by *laza*). PA is also produced from PC by PLD (encoded by *Pld*). DAG is hydrolyzed by DAG lipases, leading to the generation of PUFA. However, for PUFA generation, either an sn-2 DAG lipase or an additional enzyme (mono-acyl glycerol (MAG) lipase) would be required, but there is no evidence of either in *Drosophila* photoreceptors. Nevertheless, *sn-1* type DAG lipase (encoded by *inaE*) was isolated. This DAG lipase was predominantly localized outside the rhabdomeres. The above difficulties put in question the participation of PUFA in channel activation in vivo. Subsequently, CDP-DAG is converted into phosphatidyl inositol (PI), which is transferred back to the microvillar membrane, by the PITP (encoded by the *rdgB* gene). Both RDGA and RDGB proteins have been immunolocalized to the SMC at the base of the rhabdomere. PIP and PIP_2_ are produced at the microvillar membrane by PI kinase and PIP kinase, respectively. Bottom: The ultrastructure of highly degenerated ommatidia induced by three mutations in the genes *rdgB, cds, rdgA*, which appear in the upper scheme: and wild type for a comparison. (Modified from [43]).

**Figure 3 biomolecules-12-00382-f003:**
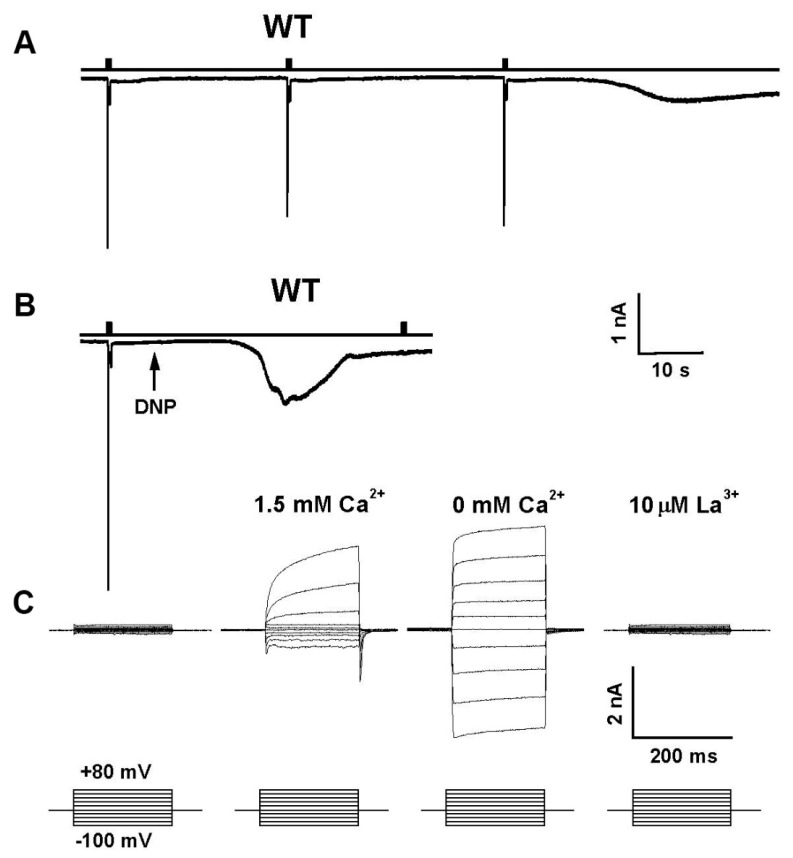
Depletion of cellular ATP activated the TRP channels: Membrane currents are usually elicited by light in the present of ATP and NAD in the pipette. Omission of these agents, combined with either few light pulses or with application of DNP to the bath induced constitutive activation of the light sensitive channels as monitored by a sustained noisy inward current that had the characteristics of the TRP or TRPL-dependent current. None of these currents were observed in the double null mutant *trpl;trp^P343^*. (**A**) The typical light induced current (LIC) of a wild type (WT) cell in response to orange lights (Schott OG 590, attenuated by 1 log unit) in the absence of ATP and NAD in the pipette. Note that after 3 light-pulses the TRP channels open spontaneously (right). The light monitor is indicated above the measured current traces. (**B**) Membrane currents were recorded 30 s after establishing the whole-cell configuration with physiological concentrations (1.5 mM) of Ca^2+^ in the bath. Whole cell recordings were conducted with pipettes, in which ATP and NAD were omitted and DNP was applied to the external medium at a time indicated by an arrow. Note the LIC was elicited by a light-pulse and the continuous opening of the channels in the dark following application of DNP. Also note that an additional light stimulus during the dark current elicited no response. (**C**): Families of current traces elicited by series of voltage steps from photoreceptors of wild type before (left traces) and following application of DNP (all other traces) in the presence of 1.5 mM Ca^2+^ in the bath (second family traces), with ~0 (nominal) mM Ca^2+^ in the bath (third family traces) and when 10 µM La^3+^ was applied to the bath (fourth family traces). For each experiment, a series of nine voltage steps was applied from a holding potential of −20 mV in 20 mV steps (bottom traces). (From [68], Copyright 2000 Society for Neuroscience).

**Figure 4 biomolecules-12-00382-f004:**
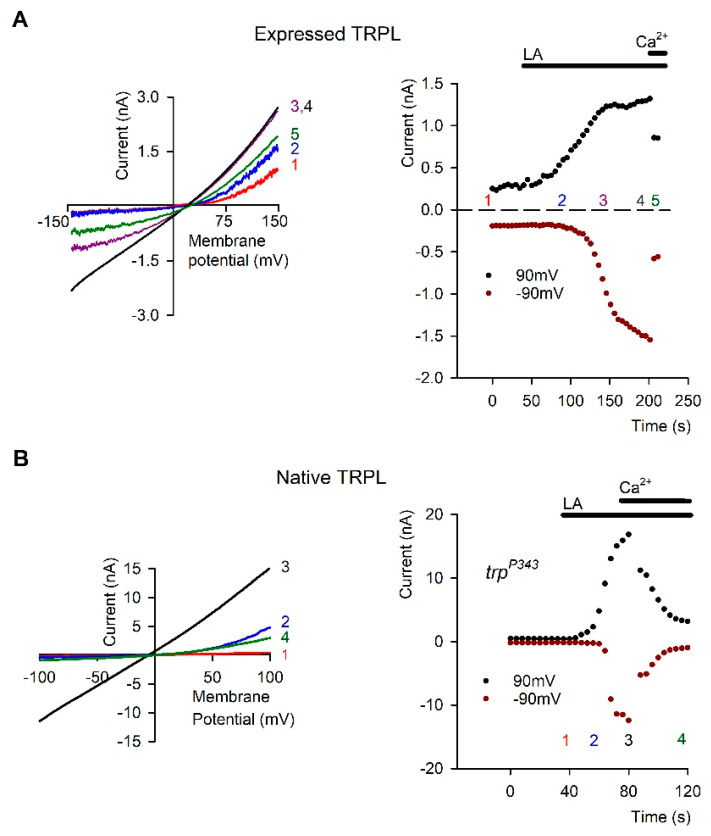
Linoleic Acid (LA) removed Open Channel Block (OCB) from TRPL Channels: (**A**) *Left*, Representative I–V curves measured from S2 cells by whole cell patch clamp recordings using voltage ramps from −150 mV to 150 mV in 1s. The typical outward rectification of the TRPL channels (red curve 1) was modified to a linear I-V curve after application of 40 µM Linoleic Acid (LA, black curve 4). The effect at positive membrane potentials preceded that of negative membrane potentials (Curves 2 and 3, blue and purple respectively, see also 1A right). Application of 5 mM Ca^2+^ restored the outward rectifying I-V curve (green curve 5). *Right*: The current values at 90 mV (black dots) and −90 mV (dark red dots) are presented as a function of time. The numbers correspond to the curves presented in the left (*n* = 15). (**B**). *Left*, Representative IV curves measured as in A from mutant *Drosophila* ommatidia that express only TRPL channels (trpP343). In darkness the TRPL channels were closed (red curve 1). After application of 60 µM LA, a linear I-V curve was obtained (black curve 3). The effect of LA at positive membrane potentials preceded the effect at negative membrane potentials in a similar manner to expressed channels (blue curve 2). Application of 10 mM Ca^2+^ blocked the TRPL channel (green curve 4, left), ruling out the possibility that the linear I-V curve was due to leak current. *Right*, the effect of LA on the current is presented as a function of time at 90 mV (black dots) and −90 mV (dark red dots). The numbers correspond to the curves presented in the left (*n* = 5). The I-V curve of the LIC is presented in pink (the maximal light intensity was attenuated by 2 log units). (From [78], Copyright 2000 Society for Neuroscience)

**Figure 5 biomolecules-12-00382-f005:**
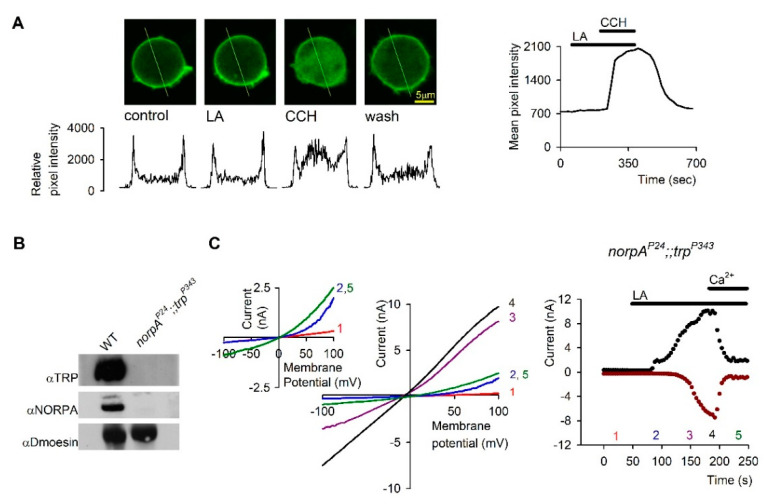
The action of Linoleic Acid (LA) is not mediated via PLC: (**A**): PLC activity was monitored by using S2 cells expressing the *Drosophila* muscarinic receptor (DM1) and eGFP-PH, which binds to PIP_2_ and IP_3_. Application of 50µM LA, under conditions which activated the channels, did not elicit any change in the eGFP-PH distribution, as monitored by confocal images of the GFP fluorescence (LA) relative to control. Application of carbachol (CCH) elicited a robust translocation of eGFP-PH to the cytosol, which was reversed by CCH removal (wash), thus indicating activation of PLC (*n* = 6). The relative fluorescence intensity at a cross section of the cell (marked by line) is also presented below the confocal images. The time course of the fluorescence changes measured in the cytosol is presented on the right. (**B**). Western blot analysis of heads homogenate of dark raised WT and *norpA^p24^;;trp^P343^* double null mutant flies. Head membrane was extracted with SDS buffer and subjected to Western blot analysis with antibodies specific for the *Drosophila* proteins TRP, NORPA and Dmoesin as indicated. No TRP and NORPA proteins were detected in the *norpA^p24^;;trp^P343^* double mutant. **(C)**. *Left*, Representative I-V curves measured by whole cell recordings from photoreceptors of the *norpA^P24^;;trp^P343^* mutant lacking PLC and the TRP channel (see Figure 5B). In darkness the TRPL channels are closed (red curve 1). The effect of the 60µM LA was not altered by the absence of PLC and a linear I-V curve was obtained (black curve 4). The effect of LA at positive membrane potentials preceded that of negative membrane potentials (Curves 2 and 3, blue and purple respectively) in a similar manner to the results of S2. Application of 10 mM Ca^2+^ restored the outward rectifying I-V curve (green curve 5). *Inset,* enlargement of curves 1, 2 and 5, demonstrating more clearly the outward rectification. *Right*, the effect of LA on the current is presented at 90mV (black dots) and −90mV (dark red dots) as a function of time. The numbers correspond to the curves presented in the left (*n* = 4). (From [78], Copyright 2000 Society for Neuroscience).

**Figure 6 biomolecules-12-00382-f006:**
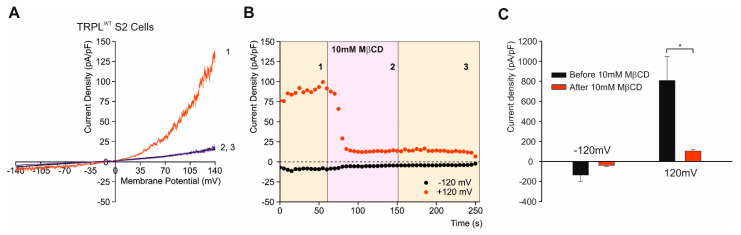
Sequestration of cholesterol abolished the constitutive activity of TRPL channels. (**A**) Current–voltage relationships (I–V curves) measuring TRPL-dependent currents I–V curves obtained in response to voltage ramp (of 1 s duration) from S2 cells expressing TRPL and showing basal channel activity with strong outward rectification, typical for TRPL-dependent current (1). The TRPL channel activity was highly reduced after perfusion with 10 mM MβCD (2) and the effect was irreversible, even after washout of MβCD (3) (*n* > 10). (**B**) Time course of the MβCD effects on TRPL currents in S2 cells. Current densities are shown as a function of time. Series of I–V curves were derived from repeatedly applied voltage ramps every 5 s and currents were measured at ± 120 mV holding potentials as a function of time under the various experimental conditions as indicated. The numbers correspond to the numbers on the I–V curves in (**A**). (**C**) Statistics of the cholesterol depletion experiments in S2 cells. Cholesterol depletion by MβCD had a significant effect on the positive TRPL currents at 120 mV (*n* = 5, values are average ± SEM, paired Student *t*-test, * *p* ≤ 0.05). (From [121]).

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
