# Peer review of "The Role of Membrane Lipids in Light-Activation of Drosophila TRP Channels"

_biomolecules, 2022, doi:10.3390/biom12030382_

Round 1

Reviewer 1 Report

This review article provides comprehensive and balanced views of current knowledge on the mechanism of regulation of Drosophila TRP and TRPL channels by membrane lipids. Both TRP and TRPL channels are involved in light-induced currents in insect’s photoreceptors. The paper is well written and it covers all the important discoveries on the relevant topic. I only noticed minor issues that require the authors’ attention.

  • Abstract line 13, “TRP channels superfamily” should be written as “TRP channel superfamily”.
  • Lines 38-43, from the way it is written, it sounds like that you define all the channels that open downstream from metabotropic receptors as metabotropic receptors too. I doubt if this is how metabotropic receptors are defined. This sentence should be revised to avoid misleading.
  • Figure 1 legend line 100, “3. The GDP to GTP exchange, in turn leads to…”. To be more consistent with the figure, it would be more accurate to state here as “3. The GTP-bound Gqα, in turn, leads to…”
  • Figure 1 legend line 104, “promote” should be “promotes” as the subject is “elevation”. “kinas” should be “kinase”.
  • Figure 1 legend line 107, “the Na-Ca2+ exchanger, CALX” should be “the Na+-Ca2+ exchanger, CALX”. Also, since this is the only place where CALX is mentioned, it would be necessary to describe in the text how CALX works at the ER to release Ca2+ in exchange for Na+ uptake. Does the Na+ concentration gradient across the ER membrane favor Na+ uptake into the ER?
  • Line 203, which diet is “This diet”? Is it the “yeast-free diet lacking PUFAs”, or the one with “the single species PUFA” added?
  • Lines 230-231, both “CDP-DAG” and “DAG-CDP” are used. It is better to keep the same style throughout unless they mean different things.
  • Lines 250-252, “Accordingly, the degeneration occurs due to …” This sentence is hard to understand. Please revise.
  • Line 257, “sever” should be “severe”.
  • Line 321 and other places, “tissue culture S2 cells” or other cells, should simply be referred to as “cultured S2 cells” and so on. There is no need to mention “tissue” in any of the places.
  • Figure 4 legend, lines 346-347. This might have been pasted here by mistake. Please check carefully.
  • Line 359, “heads homogenate” should be “head homogenates”.
  • Lines 362, 363, “;;” should be “::” (double colons instead of semicolons).
  • Line 433, “lipid modulation by a variety of poly unsaturated fatty acids cause dramatic…” should be “lipid modulation by a variety of PUFAs causes dramatic…”. “lipid modulation” is the subject. “poly unsaturated fatty acids” should be abbreviated as in all other places.
  • Line 435, “bock” should be “block”
  • Figure 6 should consider using different colors for before and after MβCD versus currents at ±120 mV, rather than limiting the colors to only orange and black.
  • Line 556, “highly value” should be “highly valued”.
  • Line 567, have the “photoactivated DAG analogues” been described in the literature? If yes, please cite the relevant paper(s).
  • Line 577, “This enzymatic reaction leads to a major change in lipid packing at the signaling membrane”. Is this just a packing issue? Would it be possible that by removing the head group, the product, DAG, is able to flip freely between the inner and outer leaflet of the lipid bilayer and bind to the DAG binding site in the channel? A DAG binding site on TRPC5 has been reported.

Author Response

  • Abstract line 13, “TRP channels superfamily” should be written as “TRP channel superfamily”.

Done.

  • Lines 38-43, from the way it is written, it sounds like that you define all the channels that open downstream from metabotropic receptors as metabotropic receptors too. I doubt if this is how metabotropic receptors are defined. This sentence should be revised to avoid misleading.

Done.

  • Figure 1 legend line 100, “3. The GDP to GTP exchange, in turn leads to…”. To be more consistent with the figure, it would be more accurate to state here as “3. The GTP-bound Gqα, in turn, leads to…”

Done.

  • Figure 1 legend line 104, “promote” should be “promotes” as the subject is “elevation”. “kinas” should be “kinase”.

Done.

  • Figure 1 legend line 107, “the Na-Ca2+exchanger, CALX” should be “the Na+-Ca2+ exchanger, CALX”. Also, since this is the only place where CALX is mentioned, it would be necessary to describe in the text how CALX works at the ER to release Ca2+ in exchange for Na+ Does the Na+ concentration gradient across the ER membrane favor Na+ uptake into the ER?

In the revised manuscript we added a paragraph describing the newly identified CALX in the ER, and mentioned that " This CALX-dependent Ca2+ release was too slow to be involved in channel activation and there is no information on ER Na+ levels. Thus the physiological role of this mechanism is still unknown."

  • Line 203, which diet is “This diet”? Is it the “yeast-free diet lacking PUFAs”, or the one with “the single species PUFA” added?

Done.

  • Lines 230-231, both “CDP-DAG” and “DAG-CDP” are used. It is better to keep the same style throughout unless they mean different things.

Done.

  • Lines 250-252, “Accordingly, the degeneration occurs due to …” This sentence is hard to understand. Please revise.

The sentence was rephrased.

  • Line 257, “sever” should be “severe”.

Done.

  • Line 321 and other places, “tissue culture S2 cells” or other cells, should simply be referred to as “cultured S2 cells” and so on. There is no need to mention “tissue” in any of the places.

Done.

  • Figure 4 legend, lines 346-347. This might have been pasted here by mistake. Please check carefully.

The sentence was deleted.

  • Line 359, “heads homogenate” should be “head homogenates”.

Done.

  • Lines 362, 363, “;;” should be “::” (double colons instead of semicolons).

This is how Drosophila mutations are annotated.

  • Line 433, “lipid modulation by a variety of poly unsaturated fatty acids cause dramatic…” should be “lipid modulation by a variety of PUFAs causesdramatic…”. “lipid modulation” is the subject. “poly unsaturated fatty acids” should be abbreviated as in all other places.

Done.

  • Line 435, “bock” should be “block”

Done.

  • Figure 6 should consider using different colors for before and after MβCD versus currents at ±120 mV, rather than limiting the colors to only orange and black.

Since this figure was already published in its present form, we prefer to leave it unchanged.

  • Line 556, “highly value” should be “highly valued”.

Done.

  • Line 567, have the “photoactivated DAG analogues” been described in the literature? If yes, please cite the relevant paper(s).

Done.

  • Line 577, “This enzymatic reaction leads to a major change in lipid packing at the signaling membrane”. Is this just a packing issue? Would it be possible that by removing the head group, the product, DAG, is able to flip freely between the inner and outer leaflet of the lipid bilayer and bind to the DAG binding site in the channel? A DAG binding site on TRPC5 has been reported.

We agree with the other possibility, deleted "major change in lipid packing…" and left the "measurable force….". in addition, we added in the section entitle- Evidence for lipids action as second messengers the following paragraph: " This hypothesis is consistent with a recent observations reporting that the human TRPC5 channels exhibit a binding site for DAG near the pore region, commensurate with a key role of DAG for TRPC5 channel activation (Song et al 2021, Storch et al 2021)".

Reviewer 2 Report

This is an excellent review article highlighting the role of lipids in the regulation of the Drosophila TRP and TRPL channels. The article is well written, concise, and will positively impact this research area. I have only one critical point to consider. In my view, the manuscript will have a more substantial impact if the authors will add a section explaining whether regulatory mechanisms of TRP and TRPL are evolutionally conserved and present in mammalian TRPCs. In addition, remarkable progress has been made in structure-functional studies with mammalian TRPCs, including cryo-EM analysis of DAG and Ca(2+) binding sites in TRPC channels, and this information is relevant for a better understanding of Drosophila TRPs.

Author Response

This is an excellent review article highlighting the role of lipids in the regulation of the Drosophila TRP and TRPL channels. The article is well written, concise, and will positively impact this research area. I have only one critical point to consider. In my view, the manuscript will have a more substantial impact if the authors will add a section explaining whether regulatory mechanisms of TRP and TRPL are evolutionally conserved and present in mammalian TRPCs. In addition, remarkable progress has been made in structure-functional studies with mammalian TRPCs, including cryo-EM analysis of DAG and Ca(2+) binding sites in TRPC channels, and this information is relevant for a better understanding of Drosophila TRPs.

The title of this review is "The role of membrane lipids in light-activation of Drosophila TRP channels". We emphasis in this review that a major evolutionally conserved mechanism is PLC mediated activation (see the beginning of the subsection: " The involvement of the inositol-lipid signaling in TRP channel activation" and "Evidence for lipids action as second messengers"). However, we think that a detailed section explaining whether regulatory mechanisms of TRP and TRPL are evolutionally conserved in mammals is outside the scope of this review. Nevertheless, we added a paragraph that mention the recent report on the DAG binding site of human TRPC5.

Reviewer 3 Report

The manuscript by Gutorov et al., is a review paper that summarizes the old and new findings of the role of membrane lipids in light-activated TRP channels. The paper is well-structured, carefully, and well written. The authors have clearly focused on the information published over the last years, which has indeed resulted in a timely review. The contribution is simple but well structured and provides interesting data on relatively unexplored aspects of TRP channels regulation.

In my opinion, this paper deserves to be published in the pages of the Biomolecules. Only minor changes are necessary:

- Line 15: Please define IP3 as you did for PLC, DAG…..

- Line 25: “In the present book chapter”. ????

- Line 38 and throughout: The reference style in the text is wrong. The references should be numbers enclosed in square brackets

- Lines 78-80: This sentence is redundant and repetitive. Please rephrase it

- Line 135: Maybe you should define “transient receptor potential” above in the text and not here

- Figure legend 2: You have already defined PLC, DAG, IP3, PIP2, PUFA…above. There is no need to repeat it.

- Line 168: “(?)”…..?????

- Line 319: “linolenic or linoleic acid (LA)”. Is it correct?

- Line 334 and throughout: Please, put a space between the number and the unit of measurement. For instance, 40uM should be 40 uM, 90mV should be 90 mV….

- The Reference list style is wrong

Author Response

Line 15: Please define IPas you did for PLC, DAG…..

Done.

- Line 25: “In the present book chapter”. ????

Done.

- Line 38 and throughout: The reference style in the text is wrong. The references should be numbers enclosed in square brackets

Done.

- Lines 78-80: This sentence is redundant and repetitive. Please rephrase it

Done.

- Line 135: Maybe you should define “transient receptor potential” above in the text and not here

Done.

- Figure legend 2: You have already defined PLC, DAG, IP3, PIP2, PUFA…above. There is no need to repeat it.

Done.

- Line 168: “(?)”…..?????

Done.

- Line 319: “linolenic or linoleic acid (LA)”. Is it correct?

Correct.

- Line 334 and throughout: Please, put a space between the number and the unit of measurement. For instance, 40uM should be 40 uM, 90mV should be 90 mV….

Done.

- The Reference list style is wrong

The reference style was fixed.
